# Resource use, availability and cost in the provision of critical care in Tanzania: a systematic review

Joseph Kazibwe [1] , Hiral A Shah [2,3] August Kuwawenaruwa [4] Carl Otto Schell [5,6] Karima Khalid,[7] Phuong Bich Tran [8] Srobana Ghosh,[9] Tim Baker [10,11] Lorna Guinness [1,3]

For numbered affiliations see end of article.

**Correspondence to**
Joseph Kazibwe;
Joseph.Kazibwe@lshtm.ac.uk

## ABSTRACT

**Objectives** Critical care is essential in saving lives of critically ill patients, however, provision of critical care across lower resource settings can be costly, fragmented and heterogenous. Despite the urgent need to scale up the provision of critical care, little is known about its availability and cost. Here, we aim to systematically review and identify reported resource use, availability and costs for the provision of critical care and the nature of critical care provision in Tanzania.

**Design** This is a systematic review following the Preferred Reporting Items for Systematic Reviews and Meta-Analyses guidelines.

**Data sources** Medline, Embase and Global Health databases were searched covering the period 2010 to 17 November 2020.

**Eligibility criteria** We included studies that reported on forms of critical care offered, critical care services offered and/or costs and resources used in the provision of care in Tanzania published from 2010.

**Data extraction and synthesis** Quality assessment of the articles and data extraction was done by two independent researchers. The Reference Case for Estimating the Costs of Global Health Services and Interventions was used to assess quality of included studies. A narrative synthesis of extracted data was conducted. Costs were adjusted and reported in 2019 US\$ and TZS using the World Bank GDP deflators.

**Results** A total 31 studies were found to fulfil the inclusion and exclusion criteria. Critical care identified in Tanzania was categorised into: intensive care unit (ICU) delivered critical care and non-ICU critical care. The availability of ICU delivered critical care was limited to urban settings whereas non-ICU critical care was found in rural and urban settings. Paediatric critical care equipment was more scarce than equipment for adults. 15 studies reported on the costs of services related to critical care yet no study reported an average or unit cost of critical care. Costs of medication, equipment (eg, oxygen, personal protective equipment), services and human resources were identified as inputs to specific critical care services in Tanzania.

**Conclusion** There is limited evidence on the resource use, availability and costs of critical care in Tanzania. There is a strong need for further empirical research on critical care resources availability, utilisation and costs across specialties and hospitals of different level in low/middle-

## STRENGTHS AND LIMITATIONS OF THIS STUDY

⇒ This study followed the Preferred Reporting Items for Systematic Reviews and Meta-Analyses guidelines to produce the first systematic review of the literature around the costs and resources used in critical care and forms of critical care in a lower resource setting such as Tanzania.

⇒ A broad and comprehensive search strategy was designed to ensure all studies on resource utilisation and costs of critical care were identified despite its heterogeneous nature.

⇒ This study used a fully articulated population, intervention, comparison and outcome framework to develop the search strategy and identify key search terms.

⇒ To address the likelihood of limited relevant data on critical care in Tanzania, we selected a set of diseases likely to require critical care (eg, malaria and pneumonia) to include in our search strategy which may have introduced some selection bias.

⇒ The availability of literature was limited in this lower-middle income country setting.

income countries like Tanzania to inform planning, priority setting and budgeting for critical care services.

**PROSPERO registration number** CRD42020221923.

## BACKGROUND

Critical care is an essential element of healthcare whose importance has been highlighted globally during the COVID-19 pandemic with countries experiencing overwhelming numbers of critically ill patients.[1–3] As a result, the care of critical illness has become an urgent point of focus in global health policy. COVID-19 notwithstanding, critical illness remains challenging driven by the large burden of injuries, infections and non-communicable diseases.[4 5] There are an estimated 45 million cases of critical illness in adults in the word each year[4] yet the limited capacity to deliver critical care, especially in resource constrained settings, is likely to continue.[6]

Critical care is the identification, monitoring and treatment of patients with critical illness through the initial and sustained support of vital organ functions.[7] It entails enhanced monitoring, treatment and attention, due to a life-threatening illness or injury.[8] There have been great advances in the provision of critical care, for example, invasive and non-invasive monitoring techniques, mechanical ventilation (MV) and renal replacement therapy, among others, resulting in reduced mortality rates in patients with critical illness, especially in high-income countries.[9]

However, such critical care is provided in sophisticated intensive care units (ICUs) using equipment which are costly to procure and maintain.[10] This makes it difficult for low resource settings to acquire such equipment to meet the needs of their populations and has led to recommendations for the development of more affordable critical care models that can be provided in any hospital settings and to all critically ill patients.[1 11 12] Moreover, most critically ill patients do not need critical care at ICU-level[13] and large unmet needs of even basic lifesaving critical care have been found in low/middle-income countries (LMICs).[14–16] With various national and international stakeholders calling for critical care scale-up,[17] it is necessary to understand the resources needed in the provision of critical care and their respective costs. Unfortunately, knowledge on the current availability of resources and the costs of critical care remains sparse.[1 18 19] There is no systematic review that examines the available resources, utilisation and costs of critical care generally[20] nor in LMIC settings.

To inform the planning and scale-up of critical care provision in a low-income country setting, this study takes a systematic approach to review available knowledge on the current resource availability, utilisation and cost in the provision of critical care in Tanzania.[20]

## METHODS

The systematic review followed the Preferred Reporting Items for Systematic Reviews and Meta-Analyses (PRISMA) guidelines[21] and the protocol was registered with PROSPERO.[20] The full methodological protocol used in this review is available elsewhere.[20] For this study, critical care is defined as the identification, monitoring and treatment of patients with critical illness through the initial and sustained support of vital organ functions.[7] It involves enhanced monitoring, treatment and attention required by a patient as a result of the life-threatening aspects of an illness or injury.[8] Some aspects of critical care can be provided in any location, for example, monitoring for vital signs and respiratory support. However, an ICU is a well-established unit intended to provide specialist critical care that sustains life of patients during periods when a patient is experiencing a life-threatening condition. ICU critical care is critical care offered in an ICU while a non-ICU critical care is critical care offered in a place that is not gazetted as an ICU.

**Table 1** PICO framework used to inform the search strategy

| Element | Description |
| --- | --- |
| Population | Any patient in need of critical care |
| Intervention | Inpatient services that could form part of critical care provision |
| Comparison | No critical care |
| Outcomes | Critical care services available<br>Cost of the different inputs to critical care provision per patient<br>Quantity and type of resources used in provision of critical care per patient |

PICO, population, intervention, comparator, outcome.

### Search strategy and selection process

Following PRISMA guidelines, three electronic databases (ie, Medline, Embase and Global Health) were systematically searched for articles published since 2010.[22] A 10-year time frame was considered to ensure that findings reflect the present-day resource utilisation paradigm as resources used and their costs vary as technology and clinical guidelines evolve. Bibliographies of included articles were reviewed to find relevant articles that fulfil the inclusion criteria but had not been identified. Google and Google Scholar were also used to search for published articles that may not have been indexed within the databases. As internet search engines typically return several thousand results, the searches were restricted to the first 50 hits and links to potentially relevant material were accessed.[23]

The PICO framework (table 1) describes major components used to identify key words. Other concepts and search terms are listed in online supplemental file 1. Key words included critical care, critical illness, resource use and setting. To ensure that no costings were missed, we further added hospital care and selected condition with high disease burden in Tanzania to the search string (eg, neonatal disorders, HIV, Malaria and tuberculosis (TB)[24]), on the premise that these are conditions that have a high risk of critical illness. See online supplemental 1 for detailed key words.

### Eligibility criteria

A study was considered eligible if it was published in English and reported on the provision of critical care,[20] services offered under critical care, resources and/or costs incurred by the health system or health providers in providing critical care in Tanzania. A detailed inclusion/exclusion criterion can be found in online supplemental 2.

### Study selection

The selection process followed the PRISMA guidelines. The articles identified during the search were retrieved and uploaded to Rayyan QCRI software.[25] The software was used to remove the duplicates. A standard process

of screening titles and abstracts and full text reading was followed in assessment of eligibility of articles. Although some papers were focused on a specific condition, those that described including the costs of critical care and/or emergency related to that condition were selected. While many diagnostics are not specific to critical care, they can be used in the delivery of critical care. We therefore also included papers that reported on diagnostic costs.

During the assessment of eligibility of the articles, two independent researchers of LG, HAS, SG, PBT and JK first reviewed the title and abstract. Articles that were identified as still eligible were then assessed through full text review by at least two researchers (SG, JK, HAS, PBT and LG). All conflicts were discussed and agreed on through consensus with third researcher (LG or HAS).

### Data extraction and synthesis

A bespoke data extraction form was developed and used to record extracted data. The extracted data included author, year of publication, context (location, setting—urban or rural, type of facility, level of facility), critical care services offered, critical care equipment available, costing perspective, costing year, currency used, type of provider, payer, source of cost data, costing time frame, direct medical costs, resources used.

A narrative synthesis of extracted data was conducted to report an overview of study characteristics of the eligible studies. We described the characteristics of critical care provision as identified in the literature and resources available for the provision of critical care in Tanzania. To facilitate comparisons, all reported costs were adjusted to 2019 prices using the GDP deflators for Tanzania[26] and reported in US$ and TZS. Cost data were collated first at the service level—by specific patient level service delivered. Then costs and resources used were categorised into groups based on a standard input-wise categorisation, for example, human resources, diagnostics, services, consumables and medication.

### Quality assessment of individual studies

The quality of the included articles was assessed by two independent researchers using an adaptation of the Reference Case for Estimating the Costs of Global Health Services and Interventions,[27] which provides a framework for quality assessment and data extraction. We used this framework to check the transparency of the authors in reporting what they did and identify likely sources of bias. With the framework, we described the following; any subgroup or population analysis done, statistical methods used to establish differences in unit costs by subgroup, determinants of cost, multivariate statistical methods used to analyse cost functions, sensitivity analyses conducted, list possible sources of bias, aspects of the cost estimates that would limit generalisability of results to other constituencies, all pecuniary and non-pecuniary interests of the study. Furthermore, methodological quality of studies that reported costs was further assessed by an appraisal checklist by Adawiyah et al.[28] We used 10 parameters to

further assess the methodological quality which included research question being tackled, perspective, time horizon, relevant inputs, methods for quantities, data sources, sample size, discount rate, sensitivity analysis and reporting of costs. Any discrepancies were addressed by a joint re-evaluation of the article among three authors.

### Patient and public involvement

This systematic review did not involve any patient nor public in the design, data extraction and analysis. However, the findings of this systematic review will be made available to the public through an open access manuscript.

## RESULTS

### Study characteristics

A total of 792 articles were found using the search and 10 articles from the reference list review. Thirty-one studies fulfilled the inclusion criteria (figure 1), of which 14 reported on the costs associated with critical care in Tanzania (please see online supplemental 3: table of study characteristics (n=31)). The disease areas covered by eligible studies included; oncology (n=1), trauma (n=4), obstetrics (n=3), neonatal/child care (n=2), HIV (n=1), renal failure (n=1), TB (n=1), malaria (n=3), stroke (n=1), typhoid (n=1), antimicrobial resistance (n=1), surgery (n=2), respiratory illness (n=1), blood transfusion (n=1), no disease or area specific (n=8). The majority of studies were cross sectional in design (n=20) and reported on the location of critical care and critical care services available.

Most studies were conducted in a specific region: the Coast Zone (n=7), Lake Zone (n=3), Northern Highland Zone (n=3), Southern Highland Zone (n=3) and Central Zone (n=1). A zone is a geographical area made up of districts within a given health region. Seven studies were nationally representative (n=7); two studies did not specify a given region. Three studies were carried out in rural settings while the rest were urban or mixed settings. Of the studies that specifically reported on critical care provision, 13 of the 15 studies were carried out in urban settings, 2 of the 15 studies were carried out in both rural and urban settings and no study was exclusively carried out in a rural setting. Nine studies specified that they were conducted in public (government owned) facilities and six specified non-governmental organisations including faith/missionary-based hospitals.

### Critical care provision in Tanzania

The structure of the publicly funded health system in Tanzania is comprised of six levels[29] starting with the village health post (lowest level), dispensaries, health centres, district hospitals, regional hospitals and tertiary referral hospitals (the highest level) and care is delivered by public, faith/missionary-based providers and private providers.[29–42] Critical care is usually offered at the hospital level only, that is, the district hospital, regional

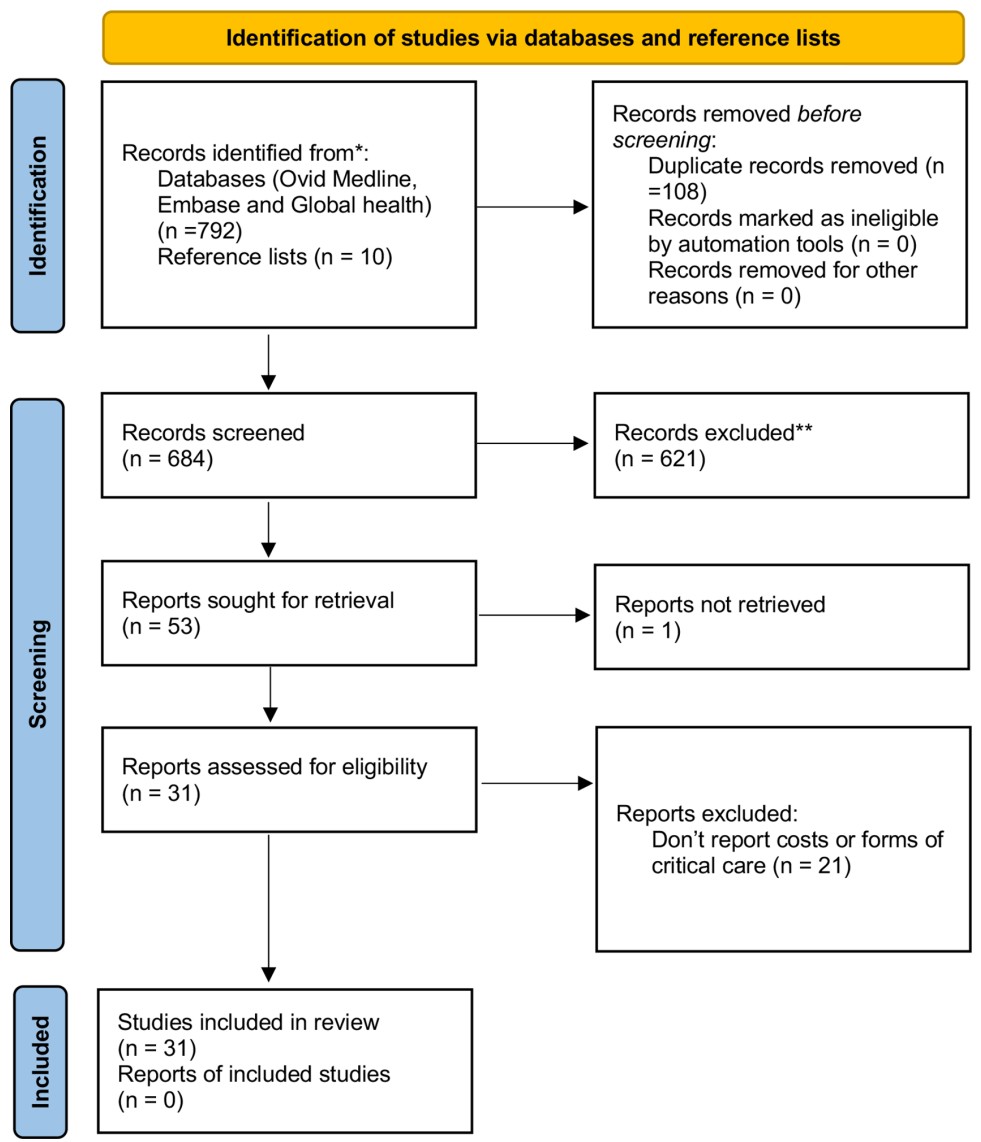

**Figure 1** PRISMA chart. The PRISMA diagram shows the details the search and selection process applied during our systematic literature search and review. Based on Page *et al*.[21] PRISMA, Preferred Reporting Items for Systematic Reviews and Meta-Analyses.

hospitals and referral hospitals.[29–34 36–43] In addition, Nicks *et al* state that district hospitals offer basic forms of critical care while referral hospitals offer critical care in ICUs.[29]

Based on the literature, we further categorise critical care into 'Critical care delivered in an ICU' and 'Critical care delivered outside the ICU'. ICU delivered critical care services included intubation and ventilation. For example, one study found that 54.2% of all ICU patients were intubated or ventilated.[41] In total, seven studies reported on ICU delivered critical care.[31 32 38 39 41 42] The ICUs were all based in tertiary referral hospitals in Tanzania including regional hospitals. For example, Murthy *et al* state that St. Francis Hospital in Ifakara region had 10 ICU beds and Sekou Toure Regional Referral Hospital in Mwanza had 8 ICU beds.[36] The district hospitals, for which studies were available, did not provide ICU delivered critical care.[37]

Critical care outside the ICU was reported to be delivered either in operating theatres or in emergency units and general wards. For example, operating theatres at Kilimanjaro Christian Medical Centre offered critical care for cases like trauma and sepsis.[44] Theatres in lower-level hospitals like district hospitals often offer initial critical care before referring the patient to higher level hospitals that can offer ICU-based critical care. Although district hospitals and some regional hospitals were reported not to have ICUs, these facilities were reported to offer critical care in the general wards.[37] Some hospitals without ICUs designated areas within general wards to cater for the critically ill patients where arterial blood gas and central venous pressure monitoring are done.[37] Administration of critical care-related medicines like Mannitol, and airway management were carried out in general wards.[42] Furthermore, Staton *et al* found that some critical

care services including intubation were performed in the emergency units.[42]

## Availability of resources required for critical care

The majority of eligible studies reported on the known resources used for critical care (n=21). However, many studies did not report the extent to which the resources were readily available at a given hospital (n=18). Three studies reported on availability and use of critical care resources and indicate that their availability varies between hospitals and is sometimes unreliable or inconsistent (online supplemental 4). Two studies find that there is a scarcity of trained staff in critical care.[37 40] Further, the availability of critical care equipment was heterogenous, with scarcity in the areas of mask and tubing, pulse oximetry, oxygen cylinders, adult and paediatric oropharyngeal airway, personal protective equipment (PPE) eye protection while gloves were abundant. In addition, a greater scarcity in paediatric than adult critical care equipment was identified.[37 40]

## Quality assessment

Of the 31 studies, 14 reported on the resource use and critical care costs and were appraised for quality using the Adawiyah et al adapted appraisal checklist[28] (online supplemental 5). All 14 studies were found to have clear research questions, mentioned the specific costing

perspective, included relevant inputs, clearly stated methods for quantifying resources and reported costs. Four studies did not specify the time horizon while four studies partially addressed it. The majority of the studies (n=8) carried out some form of sensitivity analysis. All studies that collected primary data stated sample size but did not mention how it was determined and whether it was sufficient.

## Resource use and cost of resources associated with provision of critical care

### Types of costs reported and sources of data

Studies that published costs of critical care as shown in table 2 reported economic costs (n=8) or financial costs (n=6), using a mix of bottom-up (n=9) and top-down (n=5) approaches. Sources of the costs were local data collected from hospital records, surveys and price lists. Surveys (including key informant interviews) were a common method of data collection (n=9). Three studies used patient level data on resource use to inform costs.[45–47] Hospital or laboratory records were used in 6 of the 14 studies. Some studies used both survey and records review as the source of data (n=6). Two studies used international literature as a source of data (n=2). Unit costs reported included cost per programme, cost per patient, cost per visit, cost per patient day, cost per episode, cost

| Table 2 | Types of costs reported and sources of data | | | | |
|---|---|---|---|---|---|
| **Study** | **Type of costs\*** | **Real world/ guideline cost** | **Full/incremental cost** | **Costing approach†** | **Source of cost data** |
| Githang'a et al | Financial | Real world costs | Full costs | Mixed | Hospital data |
| Mengistu et al[53] | Financial | Real world costs | Full costs | Top-down | Programme, hospital and health centre data |
| Wilson et al | Financial | Real world costs | Full costs | Bottom-up | Staff survey (n=59) |
| Kimaro et al[45] | Economic | Real world costs | Full costs | Bottom-up | Patient survey, diaries (n=870) |
| Shayo et al[52] | Economic | Real world costs | Full costs | Bottom-up | Records, government standards and market price survey |
| Phillips et al | Economic | Real world costs | Full costs | Top-down | Local programme records |
| Penno et al[51] | Economic | Real world costs | Full costs | Bottom-up | Laboratory records |
| Kabadi et al 2013 | Economic | Real world costs | Full costs | Bottom-up | Survey (n=16) and project data |
| Sicuri et al[46] | Economic | Real world costs | Full costs | Bottom-up | Patient survey (n=150) |
| Guerriero et al[50] | Economic | Real world costs | Full costs | N/A | Literature |
| Baynes et al[54] | Financial | Real world costs | Full costs | Bottom-up | Health facility records (n=31) and key informant interviews (n=124) |
| Riewpaiboon et al[47] | Economic | Real world costs | Full costs | Bottom-up | Record review and patient survey (n=17) |
| Sumona et al 2016 | Financial | Real world costs | Full costs | Bottom-up | Health facility survey (n=386) |
| James et al[48] | Financial | Real world costs | Full costs | Top-down | Facility survey (n=155) |

\*Economic costs are costs that capture opportunity, that is, the value of the forgone option/item including any donations or volunteer time, financial costs are the amount/money paid for a good or service.
†Top-down cost estimation method is the estimation of cost by dividing the total cost by the quantity of services delivered or activities carried out, bottom-up cost estimation method is based on the valuation of all resources used in the process of delivering/providing a good or service. Mixed approaches use top-down methods for certain cost categories and bottom-up for other categories where more precision is required.

**Table 3** Cost of services offered to patients

| Study | Description | Unit cost of output | Currency year | Unit cost price in 2019 (US$) | Unit cost price in 2019 (TZS) |
|---|---|---|---|---|---|
| Mengistu et al[53] | Clinical care for emergency obstetric case | Per case | 2013 | 143.85 | 329 166 |
| Mengistu et al[53] | Average cost per surgical patient | Per patient | 2013 | 285.08 | 652 333.26 |
| Mengistu et al[53] | Average cost per patient undergoing caesarean | Per patient | 2013 | 471.28 | 1 078 396.41 |
| Riewpaiboon et al[47] | Inpatient hospital cost | Per patient per day | 2010 | 91.78 | 107.49 |
| Sicuri et al[46] | Episode of malaria hospitalisation and severe anaemia | Per episode | 2009 | 57.12 | 130 704.61 |
| Sicuri et al[46] | Episode of cerebral malaria | Per episode | 2009 | 22.67 | 51 864.33 |
| Sicuri et al[46] | Episode of cerebral malaria and neurological sequela | Per episode | 2009 | 47.58 | 108 879.84 |
| Kabadi et al 2013 | Hospitalisation of stroke patient (inpatient, drugs, registration)—rural area | Per patient | 2005–2006 | 12.76 | 29 200.18 |
| Kabadi et al 2013 | Hospitalisation of stroke patient (inpatient, drugs, registration)—urban area | Per patient | 2005–2006 | 31.17 | 71 317.89 |
| Kabadi et al 2013 | Physiotherapy for stroke—urban area | Per patient | 2005–2006 | 44.88 | 102 697.20 |
| Kabadi et al 2013 | Hospitalisation of stroke patient (inpatient, drugs, registration)—average | Per patient | 2005–2006 | 18.51 | 42 361.25 |
| Kabadi et al 2013 | Physiotherapy for stroke—average | Per patient | 2005–2006 | 14.03 | 32 093.43 |
| James et al[48] | Health centre: overall | Inpatient day | 2011/2012 | 18.03 | 41 261.22 |
| James, et al[48] | Health centre: public | Inpatient day | 2011/2012 | 18.98 | 43 432.74 |
| James et al[48] | Health centre: NGO | Inpatient day | 2011/2012 | 17.29 | 39 564.68 |
| James et al[48] | Hospital: all | Inpatient day | 2011/2012 | 25.95 | 59 387.93 |
| James et al[48] | Hospital: public | Inpatient day | 2011/2012 | 18.54 | 42 422.55 |
| James et al[48] | Hospital: NGO | Inpatient day | 2011/2012 | 15.76 | 36 062.52 |
| James et al[48] | Hospital: private | Inpatient day | 2011/2012 | 68.05 | 155 721.31 |
| James et al[48] | Hospital regional | Inpatient day | 2011/2012 | 17.33 | 39 653.32 |
| James et al[48] | Hospital—ARI-pneumonia | Per case treated | 2011/2012 | 104.33 | 238 736.91 |
| James et al[48] | Hospital—hypertension | Per case treated | 2011/2012 | 253.98 | 581 166.07 |
| James et al[48] | Hospital—myocardial infarction | Per case treated | 2011/2012 | 268.57 | 614 533.21 |

per bed day, inpatient cost, total annual cost, cost per case.

### Cost of services offered to patients

Although many studies in table 3 reported costs of differing disease specific (eg, malaria and anaemia) or general hospital services (eg, inpatient and outpatient care) that included treatment of critical illness, no study reported the specific cost of the treatment of the critical care. A nationally representative study of general hospital services including critical care found that unit costs vary by provider ownership (public, non-governmental organisation (NGO) and private) and level of health facility (regional vs primary care).[48] Services provided at private

hospitals were more expensive than the same services received at public hospitals. NGO hospitals were less expensive than public hospitals.[48]

### Cost of equipment, consumables and medication

When looking at cost categories in table 4, two studies reported costs of equipment, two studies reported costs of consumables and six studies reported medication cost. Within studies, we found per item cost for equipment (4 items), consumables (3 items) and medication (13 items). The cost of a mannequin used in training was the most expensive piece of equipment reported (US$76.03).[49] In the medication category, tranexamic acid was the most

**Table 4** Cost of equipment, consumables and medication

| Study | Description | Unit cost of output | Currency year | Unit cost price in 2019 (US$) | Unit cost price in 2019 (TZS) |
|---|---|---|---|---|---|
| *Equipment* | | | | | |
| Chaudhury et al[49] | Mannequin | Per piece | 2014 | 76.03 | 173 970.65 |
| Chaudhury et al[49] | Bag-mask device | Per piece | 2014 | 16.29 | 37 279.43 |
| Chaudhury et al[49] | Penguin sucke | Per piece | 2014 | 3.26 | 7455.89 |
| Mengistu et al[53] | Equipment needed for emergency obstetric care | Per patient | 2013 | 90.02 | 205 994.65 |
| *Consumables* | | | | | |
| Mengistu et al[53] | Other supplies for emergency obstetric care | Per patient | 2013 | 28.68 | 65 635.80 |
| Mengistu et al[53] | Blood supplies | Per patient | 2013 | 0.71 | 1620.01 |
| Guerriero et al[50] | Consumables needed in the administration of tranexamic acid (10 mL syringe,100 mL bag of saline, large gauge needle) | Per dose | 2007 | 2.00 | 4576.41 |
| *Medication* | | | | | |
| Analgesics and sedatives | | | | | |
| Riewpaiboon et al[47] | Paracetamol 120 mg/5 mL in 60 mL bottle | Per bottle | 2010 | 0.33 | 750.38 |
| Riewpaiboon et al[47] | Paracetamol 500 mg tablet | Per tablet | 2010 | 0.01 | 26.80 |
| Antifungal and antibacterial drugs | | | | | |
| Riewpaiboon et al[47] | Ampicillin 500 mg vial injection | Per piece | 2010 | 0.21 | 482.39 |
| Riewpaiboon et al[47] | Ceftriaxone 250 mg vial injection | Per piece | 2010 | 0.46 | 1045.17 |
| Riewpaiboon et al[47] | Ciprofloxacin 250 mg capsule | Per capsule | 2010 | 0.01 | 26.80 |
| Riewpaiboon et al[47] | Ciprofloxacin 500 mg capsule | Per capsule | 2010 | 0.05 | 107.20 |
| Penno et al[51] | Antimicrobial treatment using generic drugs for sepsis management | Per case | 2011 | 15.17 | 34 703.66 |
| Penno et al[51] | Evidence-based antimicrobials | Per case | 2011 | 30.36 | 69 459.83 |
| Kimaro et al[45] | Cotrimoxazole per daily dose | Per daily dose | 2012 | 0.02 | 51.51 |
| Others like anti-cancers | | | | | |
| Githang'a et al 2020 | 50 mg of 6-mercaptopurine (6MP) | Unclear | 2013 | 0.39 | 885.94 |
| Mengistu et al[53] | Drugs for emergency obstetric care | per patient | 2013 | 6.90 | 15 795.12 |
| Riewpaiboon et al[47] | Glucose (5%) 500 mL | Per unit | 2010 | 0.50 | 1152.37 |
| Guerriero et al[50] | Tranexamic acid | Per dose | 2007 | 9.74 | 22 279.66 |

expensive per dose (US$9.74),[50] whereas antimicrobials were the most expensive on a per case basis (US$30.36).[51]

### Cost of diagnostics

Six studies reported on the cost of diagnostics, for example, full blood count, serum cryptococcal meningitis test, sputum culture and imaging. Lab assessment for sepsis—ID and susceptibilities positive blood culture was the most expensive diagnostic test (US$82.55) followed by CT scans (US$47.81 to 44.35) (see online supplemental 6 for details).

### Costs of human resources

Three studies reported human resource costs (see online supplemental 7). However, the breakdown of human resource costs presented in the papers did not allow for a separation of costs between treatment for the condition and the critical illness care. Reporting of human resource cost was heterogenous. One study provided monthly salaries for all personnel with monthly salaries ranging from US$2721.79 for medical specialists US$313.34 for nurses.[52] Another study reported total cost of personnel as a line item.[53] The third study reported unit costs (ie, cost per

patient) treatment for all obstetric complications broken down by staff category.[54] This study found that the highest cost was for an enrolled nurse or midwife (US$43.62 per patient) and that the lowest unit personnel cost was for an anaesthesiologist (less than US$0.01 per patient).[54]

## DISCUSSION

Our review found limited evidence on resource use, availability and costs of critical care in Tanzania. Of the eligible evidence, we found high variability in the geographical distribution and availability of resources needed for both ICU and non-ICU critical care provision. We found that ICU critical care was provided at the tertiary level in urban areas and no published provision in rural areas, whereas, non-ICU critical care was evident in both rural and urban areas. Within hospitals, we find that availability of key critical care resources (eg, pulse oximeters, oxygen cylinders, mask and tubing) can be limited but the use of these resources is not documented. We, finally, find no reported estimates on the unit costs of critical care; and that the generation and reporting of cost data is not standardised thereby limiting synthesis or aggregation.

### Availability of critical care in Tanzania

The majority of ICUs identified in our systematic review were located in tertiary referral hospitals and private hospitals located in urban areas (Dar es Salaam, Moshi, Mwanza, Mbeya city). This is of particular concern in Tanzania where the majority of the population (64.77) lives in rural areas.[55] Although this inequity in service distribution is not unique to Tanzania as previous studies in LMICs found all hospitals with ICUs were located in cities,[36] it is further exacerbated by a lack of critical care guidelines that cater to rural areas.[56] Such geographical inequity in ICU critical care could be detrimental and of particular concern during a critical care crisis (eg, COVID-19).[1]

However, critical care delivery is heterogenous and can be provided in advanced or basic forms, depending on resource availability. Critical care can involve high tech and high-cost equipment that requires highly trained staff (eg, MV) and is more often found in the ICU. Investing in this level of critical care is therefore an expensive policy decision and may explain the lack of ICU critical care availability in rural areas in Tanzania. On the other hand, investing in adequately resourced critical care outside of ICUs can potentially save lives and reduce the case load that requires ICU care.[57] We found that non-ICU critical care was reported to be given in wards and theatres[42] across rural and urban areas.

A key finding of the literature review was that studies did not provide breakdowns of costs by the level of care provided, complexity of care, resource intensity or quality of provision. Importantly, for our study, they also did not provide breakdowns between the care for a specific condition relative to the care for the critical illness. We were therefore only able to characterise critical care in

Tanzania by where the care was provided within a hospital that is, ICU versus non-ICU delivered critical care. This is of importance as care for the critically ill takes many forms/interventions depending not only on whether it is ICU-based but also by the complexity, the type of provider, the quality of care as well as the primary diagnosis. Characterising the minimum needs for the care of a critically ill patient and understanding how critical care delivery varies across these parameters will facilitate our understanding of what elements of critical care are currently available as well as recommendations for the progressive improvement of critical care in all facilities. Previous research suggests that basic critical care, which effectively identifies and treats all critically ill patients, should be available and provided in all settings within a hospital (eg, in wards) across rural and urban settings.[58 59]

Recently, some of this paper's authors have proposed the concept of Essential Emergency and Critical Care (EECC) across LMICs which could be the future of low-cost critical care. EECC is pragmatic and low-cost care of critical illness that does not need to be reliant on sophisticated technology, and have a primary focus on simple, effective actions that have large potential impact on the population[57] and the components have been specified in a large global consensus.[12] Some of the critical care interventions that have been identified as part of EECC[12] that can be provided in general wards include; identification of critical illness using vital signs-based triage, caring for blocked or threatened airway, for example, suction for secretions that are obstructing the airway and insertion of an oropharyngeal airway, caring for hypoxia or respiratory distress, threatened circulation or shock and reduced level of consciousness, see Schell *et al*[12] for further details.

Given the lack of availability and access of critical care across Tanzania, such novel approaches may provide a building block to developing a fully resourced, equitable critical care system.[60] However, decision makers require further evidence on the optimal design, costs, budget impact and cost-effectiveness of such critical care interventions in rural and urban systems.

### Complexity of resources within critical care

At the hospital level, we found that many studies identified resources that were used in the provision of services that included a critical care component but only a few studies went further to report on the availability of the resources at the facility (eg, equipment, consumables and human resources). Of the critical care components, resource availability of oxygen was explored in four studies.[37 40 42 61] Other components explored were personal protective equipment (two studies)[37 40] and availability of utilities like running water and electricity (two studies).[37 40]

While oxygen is one of the essential resources required in the provision of critical care, many hospitals still lack basic equipment for its storage and delivery.[62] Our findings show that even when oxygen is available, the equipment and consumables required to facilitate provision of oxygen therapy can be missing.[42 61] This leaves a

considerable proportion of critically ill patients in need of medical oxygen without the necessary treatment, incurring unnecessary deaths and disability.

Many reasons exist as to why availability and supply of oxygen is limited in settings such as Tanzania. For example, supply and scaling of medical oxygen is a complex process and is often controlled though monopoly suppliers. Regulation and high investment costs create barriers for new firms to enter the market and easily scale-up supply.[63] In Tanzania, medical oxygen supply is controlled centrally by a quasi-governmental monopoly organisation.[64] On the demand side, there is limited funding for medical oxygen with procurement decentralised to individual hospitals with constraint budgets.[62] Alternatives, such as oxygen concentrators and hospital plants for the local production of oxygen, are affected by the need for a constant supply of electricity, compounded by challenges with maintaining equipment. In Tanzania, similar to many LMICs, approx. 50% of hospitals are without a constant oxygen supply[40] and there is limited local capacity to maintain and repair oxygen concentrators like Tanzania.

In addition to oxygen equipment, we found variability in availability of PPE, and human resources across studies. Some PPEs especially gloves were available in most hospitals, while others, such as eye protection, remained unavailable, which is in line with previous research.[65] In terms of human resources, evidence suggests that only 20% of hospitals have at least one health worker trained in adult critical care which includes administering complex oxygen-related equipment.[37 56 58] Technical assistance, training of health workers and improved processes for non-ICU-based care, development of context-specific critical care guidelines, building capacity in equipment maintenance and biomedical engineering in facilities, and support to medical oxygen infrastructure development should be prioritised by governments and in funding applications to multilaterals and technical assistance agencies.[37 66 67]

### Costs of critical care
Costing of critical care is vital in eliciting potential budget constraints or impacts, assessing cost-effectiveness, providing insight into the economic burden of disease and can be useful for understanding the resources incurred by health systems, other payers and patients. We find that no study reported or calculated an average cost or cost per patient day for the provision of critical services. On the other hand, many studies reported individual costs of resources associated with provision of critical care and it was possible to identify the cost of specific resources needed in critical care provision (ie, human resource cost, services, medication and other consumables). Such limited evidence could be due to no formal or standardised definition of critical care existing thereby reducing specificity when costing critical care provisioning. Another possibility may be that patients who require critical care are heterogenous. The underlying demographics, epidemiology, risk factors, patient

pathway and interventions administered are also heterogenous thereby making aggregation and estimates of average cost challenging. This is further confirmed by the data extracted in this review which showed that reporting of costs within eligible studies varied by defined services provided, breakdowns of costs and inputs, as well as the cost unit, thereby making comparisons or synthesising information across studies difficult. There is an urgent need to understand the resources required and their cost implications to inform the planning of improved critical care, globally.

Almost 50% of the Tanzanian population have a daily consumption of less than US$1.91.[68] At the same time, the share of out-of-pocket payments in total health expenditure was 22% in 2019.[69] Considering the cost of high-tech critical care inputs, it is likely that critical care exposures patients to considerable out of pocket expenses resulting in catastrophic expenditure for low-income households that seek hospital care for critically ill family members. Countries should consider implementing EECC which could reduce the likely cost of critical care and therefore the financial burden on patients' households in their respective settings.

### Lessons learned for other low resource settings/LMICs
The variability in critical care provision, availability of critical care resources, prices and costs are not unique to Tanzania. The heterogeneity in critical care is well documented across the globe.[70] Variability in costs is also well documented across health facilities in LMICs.[5] The research and data gaps in these areas—understanding resource availability in critical care across different settings (both ICU and non ICU)—is less well documented but it is likely that the Tanzania's data gaps are replicated in many LMICs. LMICs will require research and data collection efforts in ascertaining these resource gaps and costing of critical care to inform budgets and planning.

### Limitations
There are a number of inherent limitations with this systematic review. First, we tailored our search strategy to account for the lack of standardised or formal definitions for critical illness, critical care provision and ICUs[58] by including specific terms for conditions likely to lead to critical illness and identified in consultation with clinicians on the authorship team. Here, we may have introduced a personal bias by focussing on specific conditions and excluding others. Second, cost and resource use in an African context are traditionally poorly documented and can fluctuate due to market dynamics and resource availability. While this systematic review provides key insights into peer-reviewed published information on the resource use, availability and costs of critical care in Tanzania, researchers must update a previous nationally representative facility survey conducted in 2012 extending it to include an understanding of the costs of critical care provision that includes patient level data on resource use. Finally, we opted to only include articles in English but believe the extent and effects of language

bias are diminished recently because of the shift towards publication of studies in English.[71]

## CONCLUSION

Critical care is essential in the prevention of mortality among critically ill patients, yet limited evidence exists on the costs and cost-effectiveness of scaling up critical care thereby hindering informed planning for investments in key life-saving interventions. We further find that ICU-based critical care provision is currently only provided in urban areas in Tanzania and there is an urgent need to assess the quality of non-ICU-based critical care implemented to address the gap in critical care provision in rural and urban areas. Evidence on resources used and potential costs in critical care is limited, and, along with a specific focus on oxygen supply and delivery which is a key resource in critical care, requires further research. Improving the evidence base on the costs, utilisation, cost-effectiveness and equity impacts of different strategies for critical care services across rural and urban areas is needed to help prevent unnecessary deaths caused by critical illness in Tanzania.

**Author affiliations**
<sup>1</sup>Global Health and Development, London School of Hygiene & Tropical Medicine, London, UK
<sup>2</sup>Department of Infectious Disease Epidemiology, Imperial College London, London, UK
<sup>3</sup>Center for Global Development, Washington, DC, USA
<sup>4</sup>Health System Impact Evaluation and Policy Unit, Ifakara Health Institute, Ifakara, United Republic of Tanzania
<sup>5</sup>Department of Global Public Health, Karolinska Institutet, Stockholm, Sweden
<sup>6</sup>Centre for Clinical Research Sörmland, Uppsala University, Eskilstuna, Sweden
<sup>7</sup>Department of Anaesthesia and Critical Care, Muhimbili University of Health and Allied Sciences, Dar es Salaam, United Republic of Tanzania
<sup>8</sup>Department of Family Medicine and Population Health, University of Antwerp, Antwerp, Belgium
<sup>9</sup>Global Health Department, Center for Global Development, Washington, DC, USA
<sup>10</sup>Department of Clinical Research, London School of Hygiene & Tropical Medicine, London, UK
<sup>11</sup>Department of Emergency Medicine, Ifakara Health Institute, Dar es Salaam, United Republic of Tanzania

**Contributors** Below is the detailed breakdown of their contribution. Concept and design: JK, HS, LG, AK, PT, OS, TB. Development of first draft: JK, HS, LGAcquisition of data: JK, HS, LG, PT, AK, KK, SG. Analysis and interpretation of data: JK, HS, LG, PT, SG, OS, KK, TB. Drafting of the manuscript: JK, HS, LG. Critical revision of paper for important intellectual content: TB, OS, LG, PT, SG, KK, AK. Statistical analysis: - Provision of study materials: Obtaining funding: TB, OS. Administrative and technical support: Supervision: LG, HS. Guarantor: JK

**Funding** This work was supported by the Wellcome Trust [221571/Z/20/Z], as part of the 'Innovation in low-and middle-income countries' Flagship.

**Competing interests** None declared.

**Patient and public involvement** Patients and/or the public were not involved in the design, or conduct, or reporting, or dissemination plans of this research.

**Patient consent for publication** Not applicable.

**Ethics approval** Ethical clearance was sought from and granted by the Tanzanian National Institute of Medical Research (reference: NIMR/HQ/R.8a/Vol. IX/3537) and London School of Hygiene and Tropical Medicine (ethics ref: 22866).

**Provenance and peer review** Not commissioned; externally peer reviewed.

**Data availability statement** Data are available upon reasonable request.

**ORCID iDs**
Joseph Kazibwe http://orcid.org/0000-0002-8315-1503
Hiral A Shah http://orcid.org/0000-0003-0204-451X
August Kuwawenaruwa http://orcid.org/0000-0001-8459-443X
Carl Otto Schell http://orcid.org/0000-0002-7904-1336
Phuong Bich Tran http://orcid.org/0000-0003-2119-998X
Tim Baker http://orcid.org/0000-0001-8727-7018
Lorna Guinness http://orcid.org/0000-0002-1013-4200

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
