## [Reviewer comments · BMJ Open]

ARTICLE DETAILS

TITLE (PROVISIONAL)	Resource use, availability and cost in the provision of critical care in Tanzania: A systematic review
AUTHORS	Kazibwe, Joseph; Shah, Hiral A.; Kuwawenaruwa, A; Schell, Carl Otto; Khalid, Karima; Tran, Phuong Bich; Ghosh, Srobana; Baker, Tim; GUINNESS, LORNA

VERSION 1 – REVIEW

REVIEWER	Jignesh Shah Bharati Vidyapeeth University
REVIEW RETURNED	09-Feb-2022

GENERAL COMMENTS	Broad search strategy was used as mentioned in methodology to retrieve data. Cost analysis won't be feasible with too much of variability amongst studies. It would have been better to conduct a survey of resource availability and do prospective (may be cross sectional) study for cost assessment across representative units to answer research question rather than systematic review. Cost variability is known across different healthcare sectors viz private and public, urban and rural. But this systematic review highlights lack of ICU resource availability and cost data across LMIC's
---

REVIEWER	Morgan Prust Yale University
REVIEW RETURNED	15-Feb-2022

GENERAL COMMENTS	This is a systematic review of critical care delivery in Tanzania, an LMIC in East Africa with a high burden of critical illness and significant resource constraints in the delivery of critical care resources. The authors have performed a nation-wide review that seeks to describe the availability of critical care resources in Tanzania and the cost of delivering them. I commend the authors' work on this important topic. The acute rise in the global critical illness burden driven by the COVID-19 pandemic, and the chronic rise driven by the global increase in non-communicable disease and population ageing ensure that global critical care delivery is and will remain a key bottleneck in global health delivery. The present study, while specific to Tanzania, has important lessons for the challenges and opportunities in developing global critical care capacity. 1) In the opening paragraph, it would be worthwhile to provide additional data about the global burden of critical illness beyond the COVID-19 pandemic. What do the authors see as the key long-term drivers of critical illness that are likely to tax underserved health care
--

	systems? 2) Critical care delivered outside the ICU is likely to be an important means of filling treatment gaps, particularly in settings with a limited number of ICU beds. The authors provide some data on critical care outside the ICU on page 12, though provide few details on the range of interventions that can be offered on a general ward. To the extent that these data are available in the studies cited, please provide more concrete examples of critical care interventions that are offered on general wards. 3) Similarly, the authors discuss non-ICU based critical care interventions in the Discussion section (pages 19-20). Please consider discussing as future direction for this work which critical care interventions can feasibly be administered in a general ward context, and how existing resources can be organized to support the delivery of these services given the resource-limitations in Tanzania and other LMIC settings. 4) The data on the cost of individual interventions is difficult to interpret as currently presented and does not give a sense of the role that cost plays in limiting access to critical care services. Without more context on who is responsible for covering these costs (individual patients/families, government payors, insurance, etc), or how prohibitive these costs are for average patients, the itemized USD costs of individual services as presented in table 4 are not particularly informative. Please consider placing these figures in the context of any existing data on mean household income in Tanzania and describing who pays these costs. 5) In the introduction, the paper is framed as a case study for critical care delivery in resource-limited settings. In the Discussion section, please consider adding a paragraph that addresses how the findings from the current study may have relevance to other LMIC contexts beyond Tanzania.
--	---

REVIEWER	Kalin Werner University of Cape Town, Division of Emergency Medicine
REVIEW RETURNED	18-May-2022

GENERAL COMMENTS	Thank you for the opportunity to review this important research to systematically report on resource use, availability, and costs of critical care delivery in Tanzania. This issue is so important and the authors work contributes to advocating for care that could save many lives. Establishing the true cost of critical care is essential to understand cost-effectiveness and budget impact of these efforts in the future. The results (or lack-thereof) will help guide both decision making, as well as research agendas. I applaud the authors for their work in this important topic. However, the manuscript could benefit from a more refined and clear definition of critical care and ICU vs non ICU care and additional details of the methodology used in the review. Foundationally the authors should provide much more detailed definitions of the terms used in their review. This is particularly important in a setting where, as the authorship team mentions, formal organization of critical care or ICU care may be limited. Critical care is a highly homogenous term for care delivered for any life-threatening injuries or illness. Throughout the manuscript it is not always clear what types of care the authors are considering critical care vs not, and it does not seem particularly useful to only consider papers which summarily cost or discuss the term “critical care intervention”. Particularly since many conditions and interventions may fall under critical care I believe it is important the authorship
---

team make a clear distinction of what is included and excluded from their definition. Perhaps it would be better for authors to indicate the type of activities considered to be critical care, or the time frame in which care is delivered? The second most important term to define in the review is the cut off for what the authors consider to be ICU vs non ICU care. Especially in a setting in which ICUs are not widely available should we only consider ICU care to be interventions delivered within a formal ICU setting? Or higher level medical interventions which may, under certain circumstances in low-resource settings with more limited health infrastructure be provided in other parts of the hospital?

Furthermore the authors should carefully review the manuscript for grammatical errors

I would like to offer the following specific suggestions below;

Abstract

- Line 57: capitalize Global Health database
- Line 59: World Bank should be capitalized

Background

- Line 100: the LMIC acronym has not been introduced yet and is later written out in full in line 103
- Line 105: What is a case study approach? Should the approach not be a systematic literature review using PRISMA guidelines?

Methods

- Line 112-113: Although the full methodological protocol is elsewhere it is important for the authors to indicate some of the methods used in their review which seem critical to the reviewers understanding of the process. For example how many reviewers screen each study? How was consensus reached? I think this would answer many subsequent questions I have in the methods section.
- Line 145: “Quantitative and qualitative syntheses” – can you expand on this? Are there any details about what these additional analyses were
- Risk bias—not clear exactly how the checklists was used? And no reporting on the outcome later on in the paper. Although the authors indicate use of this checklist the reader is not able to understand the quality of the evidence without details of the outcome of this risk bias review.

Results

- PRISMA reporting style typically involve reporting the total number of papers which were identified in the first search and the subsequent exclusions and rationale for exclusions. Could these types of details be added rather than only the final number of studies?
- In line 171-172 it is unclear how many studies reported these items and also why location is an important factor in this review about costing? It would be important for the authors to clarify their rationale for extracting this detail. Also unclear if it is necessary to identify this reporting as qualitative reporting?
- Line 174 : “and” should be removed before Southern Highland zone as it appears again afterwards in the list.
- Line 175: Perhaps clarify—are zones made up of a grouping of health regions, which are a combination of multiple districts? Currently unclear as written.
- Line 176: please write “did not” out in full rather than shortened informal didn’t.

	 • Line 177: Please remove one instance of the word “in” which is repeated twice. • Line 183: recommend changing from “comprises” to “is comprised of”. • Line 186- 188: It is a bit confusing that geographical elements of the papers are reported here when they have been recently reported in line 176-177. Appears repetitive. • Line 188-189: it would be helpful to clarify and define what you mean by ICU based critical care and non-ICU based critical care. Is this dependent on location and availability of an ICU? Or the level of care? • Line 193/ 194: This is a bit confusing. I think it would be helpful if the authors reorder this paragraph to first indicate what is meant by ICU care (e.g. services such as intubation and ventilation which require dedicated space, equipment to continue care or highly skilled health care workers) and then continue with reporting. • Line 194: please check the sentence which begins with “in addition, two...” as there is currently some verb confusion. • Line 212: Sentence which starts with “Further, the availability” seems to be a hanging/incomplete sentence—perhaps remove “highlighted”? • Resource use and cost of resources associated with provision of critical care  o It is not clear why more depth is provided on the quality appraisal checklist here for this subset of the total studies while the risk of studies is not well reported. Please provide rationale for this or add detail to the risk bias conducted. • Authors should clearly define what they mean by economic costs vs financial costs, as well as bottom up and top down approaches. • Not clear why on line 231 the authors names of the three studies are mentioned for the first time. For consistency I would recommend reporting authors and years for either all of reported outcomes or none. • Line 232-234: mentions table 2 in reference to unit costs, however these details are not reported in the table? Perhaps the authors mean table 3? • Line 238 -239: confusing and draws questions of the applicability of the data found in this review to speak to critical illness. Were the studies mentioned here excluded from the review as they did not speak about the treatment of critical illness? Authors should dive deeper in to the implications of this statement. • Line 239-241: I am not convinced that these outcomes (cost per patient for such varying conditions) are comparable. It doesn't seem particularly valuable to report these costs in this manner. Furthermore it is still confusing if the authors consider both of these reported costs as “not specific for the treatment of critical illness” as indicated in the previous sentence. And if so why are these studies included at all, and highlighted? • Line 243: “Services provided at private hospitals...” And “NGO hospitals...” important to include references here. • Line 248: Numbers under ten should be written out in full e.g, “two” • Line 254: This statement is confusing to me. How are diagnostics not a part of the provision of critical care? Isn't the condition of patients often identified with the use of diagnostics? I would argue that diagnostics are essential to guide decision making in critical care and should not be considered outside the realm of critical care delivery. In this section it would also be useful to report the range of costs your review identified in relationship to diagnostics (similar to what has been done above for equipment, consumables, medication).
--	--

	 • Line 255: need to write FBC in full. • Line 258: “No human resource costs...”. This is unclear. As I understand the authors have determined which studies are considered ICU or non ICU (as per line 189). Thus can the authors used this to identify which studies include human resources costs in each category and report these details? • Line 260- 262: Please at references for each of the statements so the reader is aware of which studies these statements refer to. Discussion:  • Line 292: Could the authors clarify what they mean here? Does this mean none of the details in this list were reported in any of the studies? • Line 294: it’s unclear what the authors mean here, aren’t some of the studies in this review about care delivered outside of the ICU? Don’t the authors report on these interventions outside of ICU earlier in the paper? • Line 295- 297 : need to clarify as this is making a different point that the prior sentence. Is the issue that there is not an understanding of minimum needs? Or that there is not enough evidence of what elements of critical care are currently available? These are two separate issues, and the prior could use some references. • Line 300: Would be useful to share some clarifying examples of what EECC interventions are • Line 311: Should either say “but few studies” or “but only a few” to indicate scarcity • Line 319/320: References? • Line 318-325: This paragraph on oxygen seems out of place. Upon second review I notice many studies included in the review speak to oxygen deliver which is perhaps why the authorship team has chosen to spend a significant portion of the discussion on this singular intervention. However that should be clarified. As it currently stands it seems that the overall manuscript is about critical care delivery (quite broad) yet most of the discussion is dedicated to a single intervention (oxygen delivery). • Line 327: “while some personal...” please check this sentence as it seems to run on. • Line 333: Interesting the authors indicate multilaterals and technical assistance agencies for these priorities. Why not government or decision makers? • Line 346: Unclear here what the authors mean by “eligible studies were inconsistent”. How so? Can detail be provided for why it is not possible to compare or synthesise these results?
--	--

VERSION 1 – AUTHOR RESPONSE

#	Reviewer #1 s’ comments	Authors’ response
1	Broad search strategy was used as mentioned in methodology to retrieve data. Cost analysis won't be feasible with too much of variability amongst studies. It would have been better to conduct a survey of resource availability and do prospective (may be cross sectional) study for cost assessment across representative	Thank you for the comment. We agree with the reviewer that ideally, we would carry out a survey of resource availability as well as prospective cost data and we hope that these will be feasible in the near future. The systematic review provides a preparatory step towards this end – the motivation is to identify the level of data that already exists in this area and the relevant gaps in research and evidence to inform future

	units to answer research question rather than systematic review. Cost variability is known across different healthcare sectors viz private and public, urban and rural. But this systematic review highlights lack of ICU resource availability and cost data across LMIC's	survey design. We also agree that variability in the costs is a significant problem, particularly in the area of critical care. We agree with the reviewer that our study highlights the critical care data limitations that exist in LMICs setting like Tanzania as stated on Page 25 lines 839-851.
#	Reviewer #2 s' comments	Authors' response
1	In the opening paragraph, it would be worthwhile to provide additional data about the global burden of critical illness beyond the COVID-19 pandemic. What do the authors see as the key long-term drivers of critical illness that are likely to tax underserved health care systems?	Thank you for the suggestion. We have included the following information in the first paragraph. “COVID-19 notwithstanding, critical illness remains challenging driven by the large burden of injuries, infections and noncommunicable diseases ^{4,5}. There are an estimated 45 million cases of critical illness in adults in the world each year⁴ yet the limited capacity to deliver critical care, especially in resource constrained settings, is likely to continue ⁶.” Page 5 Line 136-139
2	Critical care delivered outside the ICU is likely to be an important means of filling treatment gaps, particularly in settings with a limited number of ICU beds. The authors provide some data on critical care outside the ICU on page 12, though provide few details on the range of interventions that can be offered on a general ward. To the extent that these data are available in the studies cited, please provide more concrete examples of critical care interventions that are offered on general wards.	Thank you for the suggestion. We have addressed it by including the following “.....Some hospitals without ICUs designated areas within general wards to cater for the critically ill patients where arterial blood gas and central venous pressure monitoring are done³⁸. Administration of critical care related medicines like Mannitol, and airway management were carried out in general wards.⁴³” Page 13 lines 385-388 In addition we have discussed some of the interventions that can be carried out in a general ward in the following statement, “Some of the critical care interventions that have been identified as part of EECC¹² that can be provided in general wards include; identification of critical

		illness using vital signs based triage, caring for blocked or threatened airway for example suction for secretions that are obstructing the airway and insertion of an oropharyngeal airway, caring for hypoxia or respiratory distress, threatened circulation or shock and reduced level of consciousness, see Schell et al¹² for further details.” Page 21 Lines 737-742
3	Similarly, the authors discuss non-ICU based critical care interventions in the Discussion section (pages 19-20). Please consider discussing as future direction for this work which critical care interventions can feasibly be administered in a general ward context, and how existing resources can be organized to support the delivery of these services given the resource-limitations in Tanzania and other LMIC settings.	Thanks for comment; we have included the following, “of Essential Emergency and Critical Care (EECC) across LMICs which could be the future of low-cost critical care. EECC is pragmatic and low-cost care of critical illness that does not need to be reliant on sophisticated technology, and have a primary focus on simple, effective actions that have large potential impact on the population⁵⁹ and the components of have been specified in a large global consensus¹². Some of the critical care interventions that have been identified as part of EECC¹² that can be provided in general wards include; identification of critical illness using vital signs based triage, caring for blocked or threatened airway for example suction for secretions that are obstructing the airway and insertion of an oropharyngeal airway, caring for hypoxia or respiratory distress, threatened circulation or shock and reduced level of consciousness, see Schell et al¹² for further details.” Page 21 lines 732 -742
4	The data on the cost of individual interventions is difficult to interpret as currently presented and does not give a sense of the role that cost plays in limiting access to critical care services. Without more context on who is responsible for covering these costs (individual patients/families, government payors, insurance, etc), or how prohibitive these costs are for average patients, the itemized USD costs of individual services as presented in table 4 are not particularly informative. Please consider placing these figures in the context of any	Thanks for the comment, we have included the following, “Almost 50% of the Tanzanian population have a daily consumption of less than 1.91USD⁷⁰. At the same time, the share of out-of-pocket payments in total health expenditure was 22% in 2019⁷¹. Considering the cost of high-tech critical care inputs, it is likely that critical care exposures patients to considerable out of pocket expenses resulting in catastrophic expenditure for low-income households that seek hospital care for critically ill family members. Countries should consider implementing EECC which could reduce the likely cost of critical care and

	existing data on mean household income in Tanzania and describing who pays these costs.	therefore the financial burden on patients' households in their respective settings.” Page 24 lines 824-830
5	In the introduction, the paper is framed as a case study for critical care delivery in resource-limited settings. In the Discussion section, please consider adding a paragraph that addresses how the findings from the current study may have relevance to other LMIC contexts beyond Tanzania.	Thank you for the suggestion. We agree. We have aligned the recommendations to fit the LMIC context and provided further references to support the discussion including a sub-section summarizing the lessons learned for elsewhere: “Lessons learned for other low resource settings/LMICs The variability in critical care provision, availability of critical care resources, prices and costs are not unique to Tanzania. The heterogeneity in critical care is well documented across the globe⁷². Variability in costs is also well documented across health facilities in LMICs⁵. The research and data gaps in these areas— understanding resource availability in critical care across different settings (both ICU and non ICU) - is less well documented but it is likely that the Tanzania’s data gaps are replicated in many LMICs. LMICs will require research and data collection efforts in ascertaining these resource gaps and costing of critical care to inform budgets and planning.” Pages 24&25 Lines 832-850 “..Alternatives, such as oxygen concentrators and hospital plants for the local production of oxygen, are affected by the need for a constant supply of electricity, compounded by challenges with maintaining equipment. In Tanzania, similar to many LMICs, approx. 50% of hospitals are without a constant oxygen supply⁴¹ and there”

		Page 22 lines 775-786
#	Reviewer #3 s' comments	Authors' response
1	However, the manuscript could benefit from a more refined and clear definition of critical care and ICU vs non ICU care and additional details of the methodology used in the review.	Thank you for the comment. We have updated our definition of critical care in the methods section as follows: “For this study, critical care is defined as the identification, monitoring and treatment of patients with critical illness through the initial and sustained support of vital organ functions⁷. It involves enhanced monitoring, treatment and attention required by a patient as a result of the life-threatening aspects of an illness or injury⁸. Some aspects of critical care can be provided in any location e.g. monitoring for vital signs and respiratory support. However, an ICU is a well-established unit intended to provide specialist critical care that sustains life of patients during periods when a patient is experiencing a life-threatening condition. ICU critical care is critical care offered in an ICU while a non-ICU critical care is critical care offered in a place that is not gazetted as an ICU.” Page 6&7 lines 183-200 We have also provided further examples of where critical care is delivered in a hospital in the results section under the subsection “Critical care in Tanzania”. Page 12
2	Foundationally the authors should provide much more detailed definitions of the terms used in their review. This is particularly important in a setting where, as the	Thank you for the comment. We have further refined our operational definitions for the terms you have raised (see

authorship team mentions, formal organization of critical care or ICU care may be limited. Critical care is a highly homogenous term for care delivered for any life-threatening injuries or illness. Throughout the manuscript it is not always clear what types of care the authors are considering critical care vs not, and it does not seem particularly useful to only consider papers which summarily cost or discuss the term “critical care intervention”. Particularly since many conditions and interventions may fall under critical care I believe it is important the authorship team make a clear distinction of what is included and excluded from their definition. Perhaps it would be better for authors to indicate the type of activities considered to be critical care, or the time frame in which care is delivered? The second most important term to define in the review is the cut off for what the authors consider to be ICU vs non ICU care. Especially in a setting in which ICUs are not widely available should we only consider ICU care to be interventions delivered within a formal ICU setting? Or higher-level medical interventions which may, under certain circumstances in low-resource settings with more limited health infrastructure be provided in other parts of the hospital?	above) We have also added detail to search strategy in relation to the search terms: “Key words included critical care, critical illness, resource use and setting. To ensure that no costings were missed, we further added hospital care and selected condition with high disease burden in Tanzania to the search string (for example neonatal disorders, HIV, Malaria and TB²³), Page 8 Lines 218- 221 And provided further information on the selection process: “Study selection The selection process followed the PRISMA guidelines. The articles identified during the search were retrieved and uploaded to Rayyan QCRI software²⁶. The software was used to remove the duplicates. A standard process of screening titles and abstracts and full text reading was followed in assessment of eligibility of articles. Although some papers were focussed on a specific condition, those that described including the costs of critical care and/or emergency related to that condition were selected. While many diagnostics are not specific to critical care, they can be used in the delivery of critical care. We therefore also included papers that reported on diagnostic costs. During the assessment of eligibility of the articles, two independent researchers of LG, HAS, SG, PBT and JK first reviewed the title and abstract. Articles that were identified as still eligible were then assessed through full text review by at least two researchers (SG, JK, HAS, PBT and LG). All conflicts were discussed and agreed on through consensus with third
---	---

		researcher (LG or HAS).” Page 8&9 Lines 229-247
	Minor comments	
	Abstract	
3	Line 57: capitalize Global Health database	Changes have been made “... Embase and Global Health databases” Page 2 Line 67
4	Line 59: World Bank should be capitalized	Changes have been made as below, “....TZS using the World Bank GDP deflators.” Page 2 Lines 70
	Background	
5	Line 100: the LMIC acronym has not been introduced yet and is later written out in full in line 103	Changes have been made as below, “... lifesaving critical care have been found in low and middle income countries (LMICs) ¹²⁻¹⁴ . With various national and international stakeholders calling for critical care scale-up ¹⁵ , it is necessary to understand the resources needed in the provision of critical care and their respective costs. Unfortunately, knowledge on the current availability of resources and the costs of critical care remains sparse ^{1,16,17} . There is no systematic review that examines the available resources, utilisation and costs of critical care generally ¹⁸ nor in LMIC settings.” Page 5&6 Line 153-171
6	Line 105: What is a case study approach? Should the approach not be a systematic literature review using PRISMA guidelines?	Thank you for the question. We have made the following change, “To inform the planning and scale up of critical care provision in a low-income country setting, this study takes a systematic approach to review available knowledge on the current resource availability, utilisation and cost in the provision of critical care in Tanzania”

		Page 6 Lines 174
	Methods	
7	Line 112-113: Although the full methodological protocol is elsewhere it is important for the authors to indicate some of the methods used in their review which seem critical to the reviewers understanding of the process. For example how many reviewers screen each study? How was consensus reached? I think this would answer many subsequent questions I have I the methods section.	Thank you for the suggestion. We have included the requested information as below, “Study selection The selection process followed the PRISMA guidelines. The articles identified during the search were retrieved and uploaded to Rayyan QCRI software²⁶. The software was used to remove the duplicates. A standard process of screening titles and abstracts and full text reading was followed in assessment of eligibility of articles. Although some papers were focussed on a specific condition, those that described including the costs of critical care and/or emergency related to that condition were selected. While many diagnostics are not specific to critical care, they can be used in the delivery of critical care. We therefore also included papers that reported on diagnostic costs. During the assessment of eligibility of the articles, two independent researchers of LG, HAS, SG, PBT and JK first reviewed the title and abstract. Articles that were identified as still eligible were then assessed through full text review by at least two researchers (SG, JK, HAS, PBT and LG). All conflicts were discussed and agreed on through consensus with third researcher (LG or HAS).” Page 8&9 Lines 229-247
8	Line 145: “Quantitative and qualitative syntheses” – can you expand on this? Are there any details about what these additional analyses were	Thank you for this input. As the analysis is descriptive for the cost and resource use data, we have changed the description of the analysis to: “To facilitate comparisons, all reported costs were adjusted to 2019 prices using the GDP deflators for Tanzania²⁷ and reported in USD and TZS. Cost data was collated first at the service level - by specific patient level service delivered. Then costs and resources used were categorized into groups based on a standard input-wise categorisation, for example, human resources,

		diagnostics, services, consumables and medication.” Page 9 Lines 257-262
9	Risk bias—not clear exactly how the checklists was used? And no reporting on the outcome later on in the paper. Although the authors indicate use of this checklist the reader is not able to understand the quality of the evidence without details of the outcome of this risk bias review.	Thanks for the suggestion. We have provided more details about the checklist as below; “The quality of the included articles was assessed by two independent researchers using an adaptation of the Reference Case for Estimating the Costs of Global Health Services and Interventions²⁸, which provides a framework for quality assessment and data extraction. We used this framework to check the transparency of the authors in reporting what they did and identify likely sources of bias. With the framework we described the following; any sub group or population analysis done, statistical methods used to establish differences in unit costs by sub-group, determinants of cost, multivariate statistical methods used to analyze cost functions, sensitivity analyses conducted, list possible sources of bias, aspects of the cost estimates that would limit generalizability of results to other constituencies, all pecuniary and non-pecuniary interests of the study. Furthermore, methodological quality of studies that reported costs were further assessed by an appraisal checklist by Adawiyah et al²⁹. We used ten parameters to further assess the methodological quality which included research question being tackled, perspective, time horizon, relevant inputs, methods for quantities, data sources, sample size, discount rate, sensitivity analysis, and reporting of costs. Any discrepancies were addressed by a joint re-evaluation of the article among 3 authors.” Page 10 Lines 279-293 In the results section we have created a subsection entitled “quality assessment” which contains the quality assessment results. This is showed below;

		“Quality assessment Of the 31 studies, 14 reported on the resource use and critical care costs and were appraised for quality using the Adawiyah et al adapted appraisal checklist²⁹ (Supplementary 5). All 14 studies were found to have clear research questions, mentioned the specific costing perspective, included relevant inputs, clearly stated methods for quantifying resources and reported costs. Four studies did not specify the time horizon while four studies partially addressed it. The majority of the studies (n=8) carried out some form of sensitivity analysis. All studies that collected primary data stated sample size but did not mention how it was determined and whether it was sufficient.” Page 13&14 Lines 402-413
	Results	
10	PRISMA reporting style typically involve reporting the total number of papers which were identified in the first search and the subsequent exclusions and rationale for exclusions. Could these types of details be added rather than only the final number of studies?	Thanks for the suggestion. We had included the details in the PRISMA flow chart (figure 1). In addition, we have included the following in text, “Study characteristics A total of 792 articles were found using the search and 10 articles from the reference list review. Thirty-one studies fulfilled the inclusion criteria (figure 1),” Page 11, Line 307-308
11	In line 171-172 it is unclear how many studies reported these items and also why location is an important factor in this review about costing? It would be important for the authors to clarify their rationale for extracting this detail. Also unclear if it is necessary to identify this reporting as qualitative reporting?	Thank you for the question. The location where critical care is offered is important because it has implications on the items/resources used and thereby the cost of providing the care. That's why we included it.

		We agree with the reviewer about qualitative reporting. We have made modifications as below, “... A narrative synthesis of extracted data was conducted to report an overview of study characteristics of the eligible studies. We described the characteristics of critical care provision as identified in the literature and resources available for the provision of critical care in Tanzania.” Page 09 Lines 255-257
12	Line 174 : “and” should be removed before Southern Highland zone as it appears again afterwards in the list.	Agreed and modification made. Page 11 Line 317
13	Line 175: Perhaps clarify—are zones made up of a grouping of health regions, which are a combination of multiple districts? Currently unclear as written.	We have rewritten it as follows, “A zone is a geographical area made up of districts within a given health region” Page 11 Line 318
14	Line 176: please write “did not” out in full rather than shortened informal didn’t.	Agreed and modification made. “... two studies did not specify a given region.” Page 11 Line 319
15	Line 177: Please remove one instance of the word “in” which is repeated twice.	Agreed and modification made. Page 11 Line 324
16	Line 183: recommend changing from “comprises” to “is comprised of”.	Agreed, we have modified the sentence as follows, “The structure of the health system in Tanzania is comprised of six levels²⁸ starting with the village

		health post (lowest level),...” Page 12 lines 340
17	Line 186- 188: It is a bit confusing that geographical elements of the papers are reported here when they have been recently reported in line 176-177. Appears repetitive.	Thank you for this observation. We have clarified this by changing the phrasing and specifying that those studies reporting on ICU care were all in ICUs located in tertiary level hospitals. “Based on the literature, we further categorise critical care into “Critical care delivered in an ICU” and “Critical care delivered outside the ICU”. ICU delivered critical care services included intubation and ventilation. For example, one study found that 54.2% of all ICU patients were intubated or ventilated⁴². In total, seven studies reported on ICU delivered critical care ^{32,33,39,40,42,43}.” Page 12 lines 348-352 In addition, we have included a sub-heading “Critical care provision in Tanzania” to help with the distinction Page 12 line 339
18	Line 188-189: it would be helpful to clarify and define what you mean by ICU based critical care and non-ICU based critical care. Is this dependent on location and availability of an ICU? Or the level of care?	Thank you for the suggestion. We have tried to clarify this by changing the phrasing and specifying that those studies reporting on ICU care were all in ICUs located in tertiary level hospitals. “Based on the literature, we further categorise critical care into “Critical care delivered in an ICU” and “Critical care delivered outside the ICU”. ICU delivered critical care services included intubation and ventilation. For example, one study found that 54.2% of all ICU patients were intubated or

		ventilated⁴². In addition, two studies reported In total, seven studies reported on ICU delivered critical care ^{32,33,39,40,42,43}. The ICUs were all based in tertiary referral hospitals in Tanzania including regional hospitals.” Pages 12 Lines 348-353
19	Line 193/ 194: This is a bit confusing. I think it would be helpful if the authors reorder this paragraph to first indicate what is meant by ICU care (e.g. services such as intubation and ventilation which require dedicated space, equipment to continue care or highly skilled health care workers) and then continue with reporting.	Thank you for the suggestion. We have re-ordered the paragraph in line with your recommendation. Pages 12 Lines 348-353
20	Line 194: please check the sentence which begins with “in addition, two...” as there is currently some verb confusion.	Agreed, we have removed the sentence.
21	Line 212: Sentence which starts with “Further, the availability” seems to be a hanging/incomplete sentence—perhaps remove “highlighted”?	Agreed, modifications effected. “Further, the availability of critical care equipment was heterogenous, with scarcity in the areas of mask and tubing, pulse oximetry, oxygen cylinders, adult and paediatric oropharyngeal airway, PPE eye protection while gloves were abundant.” Page 13 Lines 398-400
22	Resource use and cost of resources associated with provision of critical care It is not clear why more depth is provided on the quality appraisal checklist here for this subset of the total studies while the risk of studies is not well reported. Please provide rationale for this or add detail to the risk bias conducted.	Agreed. We have created a subsection for the quality assessment and moved the text accordingly as below, “Quality assessment Of the 31 studies, 14 reported on the resource use and critical care costs and were appraised for quality using the Adawiyah et al adapted appraisal checklist²⁸ (Supplementary 5). All 14 studies were found to have clear research questions, mentioned the specific costing

		perspective, included relevant inputs, clearly stated methods for quantifying resources and reported costs. Four studies did not specify the time horizon while four studies partially addressed it. The majority of the studies (n=8) carried out some form of sensitivity analysis. All studies that collected primary data stated sample size but did not mention how it was determined and whether it was sufficient.” Page 13&14 Lines 402-413
23	Authors should clearly define what they mean by economic costs vs financial costs, as well as bottom up and top down approaches.	Agreed, we have included the definitions as follows in table 2; “*Economic costs are costs that capture opportunity i.e. the value of the forgone option/item including any donations or volunteer time, financial costs are the amount/money paid for a good or service. **Top down cost estimation method is the estimation of cost by dividing the total cost by the quantity of services delivered or activities carried out, bottom up cost estimation method is based on the valuation of all resources used in the process of delivering/providing a good or service. Mixed approaches use top down methods for certain cost categories and bottom up for other categories where more precision is required.” Page 15 lines 451-456
24	Not clear why on line 231 the authors names of the three studies are mentioned for the first time. For consistency I would recommend reporting authors and years for either all of reported outcomes or none.	Agreed, the names have been deleted. Page 14 Line 423
25	Line 232-234: mentions table 2 in reference to unit costs, however these details are not reported in the table? Perhaps the authors mean table 3?	Agreed, table 2 was modified and column for unit cost removed because it was going to be reported in subsequent tables.

		The mention of “table 2” has been removed. Page 14 Line 427
26	Line 238 -239: confusing and draws questions of the applicability of the data found in this review to speak to critical illness. Were the studies mentioned here excluded from the review as they did not speak about the treatment of critical illness? Authors should dive deeper in to the implications of this statement.	Thank you so much for the comment. We had wanted to communicate that no study reported the comprehensive cost of critical care treatment. We agree that that part of the sentence may be confusing to the reader and has therefore been modified to, “Although many studies reported costs of differing disease specific (e.g. malaria and anaemia) or general hospital services (e.g. inpatient and outpatient care) that included treatment of critical illness, no study reported the specific cost of the treatment of that critical care”. Page 17 lines 615-617
27	Line 239-241: I am not convinced that these outcomes (cost per patient for such varying conditions) are comparable. It doesn’t seem particularly valuable to report these costs in this manner. Furthermore it is still confusing if the authors consider both of these reported costs as “not specific for the treatment of critical illness” as indicated in the previous sentence. And if so why are these studies included at all, and highlighted?	Thank you for the comment. We agree that highlighting these particular findings is not necessary and does not add to the paper. We have deleted these sentences. Page 17 line 617
28	Line 243: “Services provided at private hospitals....” And “NGO hospitals...” important to include references here.	Reference has been provided. “A nationally representative study of general hospital services including critical care found that unit costs vary by provider ownership (public, NGO and private) and level of health facility (regional vs primary care) ⁴⁹. Services provided at private hospitals were more expensive than the same services received at public hospitals. NGO hospitals were less expensive than public hospitals⁴⁹.”

		Page 17 lines 619-621
29	Line 248: Numbers under ten should be written out in full e.g, “two”	Agreed, the digits have been written in words as below, “When looking at cost categories, two studies reported costs of equipment, two studies reported costs of consumables and six studies reported medication cost.” Page 18 lines 638-639
30	Line 254: This statement is confusing to me. How are diagnostics not a part of the provision of critical care? Isn't the condition of patients often identified with the use of diagnostics? I would argue that diagnostics are essential to guide decision making in critical care and should not be considered outside the realm of critical care delivery. In this section it would also be useful to report the range of costs your review identified in relationship to diagnostics (similar to what has been done above for equipment, consumables, medication).	We acknowledge this comment and have deleted the first sentence of this paragraph. We have added a table of results related to the diagnostics to the supplementary material (see supplementary 6). Further we have included a sentence that summarises the cost ranges for the diagnostics: “Lab assessment for sepsis- ID and susceptibilities positive blood culture was the most expensive diagnostic test (USD 82.55) followed by CT scans (USD 47.81 to 44.35). (see supplementary 6 for details).” Page 18 lines 646-648
31	Line 255: need to write FBC in full.	We have made modifications as below, “Six studies reported on the cost of diagnostics, for example, full blood count (FBC), serum cryptococcal meningitis test, sputum culture and imaging.” Page 18 Line 645
32	Line 258: “No human resource costs...”. This is unclear. As I understand the authors have determined which studies are considered ICU or non ICU (as per line 189). Thus can the authors used this to identify which studies include human resources costs in each	Thank you for this comment. We have made edits as below to clarify this, “However, the breakdown of human resource costs presented in the papers did not allow for a separation of costs between treatment for the

	category and report these details?	condition and the critical illness care.” Page 18 Lines 650-652
33	Line 260- 262: Please at references for each of the statements so the reader is aware of which studies these statements refer to.	We have included the references as shown below, “Another study reported total cost of personnel as a line item ⁵⁴ . The third study reported unit costs (i.e. cost per patient) treatment for all obstetric complications broken down by staff category ⁵⁵ .” Pages 18 lines 654-655
	Discussion	
34	Line 292: Could the authors clarify what they mean here? Does this mean none of the details in this list were reported in any of the studies?	Thanks for the comment. Here we meant that no study holistically reported on those items. We have modified it as below, “A key finding of the literature review was that studies did not provide breakdowns of costs by the level of care provided, complexity of care, resource intensity or quality of provision. Importantly, for our study, nor did they provide breakdowns between the care for a specific condition relative to the care for the critical illness.” Page 20 lines 705-708
35	Line 294: it's unclear what the authors mean here, aren't some of the studies in this review about care delivered outside of the ICU? Don't the authors report on these interventions outside of ICU earlier in the paper?	Yes, there are studies that report critical care interventions offered outside the ICU. Such studies create the base for why we were able to classify critical care based on the point where it is offered (ICU vs non-ICU). We have tried to clarify this point by changing the wording as follows: “We were therefore only able to characterise critical care in Tanzania by where the care was provided within a hospital i.e. ICU vs non-ICU delivered critical care. This is of importance as care for the critically ill takes many forms/interventions and is.....”

		Page 20 lines 708-711
36	Line 295- 297 : need to clarify as this is making a different point that the prior sentence. Is the issue that there is not an understanding of minimum needs? Or that there is not enough evidence of what elements of critical care are currently available? These are two separate issues, and the prior could use some references.	Thank you for this. We believe the changes above help clarify the issue here and have further edited the subsequent sentences: “This is of importance as care for the critically ill takes many forms/interventions depending not only on whether it is ICU based but also by the complexity, the type of provider, the quality of care as well as the primary diagnosis. Characterising the minimum needs for the care of a critically ill patient and understanding how critical care delivery varies across these parameters will facilitate our understanding of what elements of critical care are currently available as well as recommendations for the progressive improvement of critical care in all facilities.” Page 20 lines 710-716
37	Line 300: Would be useful to share some clarifying examples of what EECC interventions are	Thank you for the suggestion. We have added it as below, “Some of the critical care interventions that have been identified as part of EECC¹⁰ that can be provided in general wards include; identification of critical illness using vital signs based triage, caring for blocked or threatened airway for example suction for secretions that are obstructing the airway and insertion of an oropharyngeal airway, caring for hypoxia or respiratory distress, threatened circulation or shock and reduced level of consciousness, see Schell et al¹² for further details. ” Pages 21 Lines 737-742
38	Line 311: Should either say “but few studies” or “but only a few” to indicate scarcity	Agreed. We have made the change as follows, “At the hospital level, we found that many studies identified resources that were used in the provision of services that included a critical care component but only a few studies went further”

		Page 22 Lines 755-757
39	Line 319/320: References?	Reference included, “Regulation and high investment costs create barriers for new firms to enter the market and easily scale up supply ⁶⁴ . ” Page 22 lines 771-772
40	Line 318-325: This paragraph on oxygen seems out of place. Upon second review I notice many studies included in the review speak to oxygen deliver which is perhaps why the authorship team has chosen to spend a significant portion of the discussion on this singular intervention. However that should be clarified. As it currently stands it seems that the overall manuscript is about critical care delivery (quite broad) yet most of the discussion is dedicated to a single intervention (oxygen delivery).	This is a good point; we have clarified by adding detail on the number of studies that look at oxygen (and other components) as follows: “Of the critical care components, resource availability of oxygen was explored in four studies ^{38,41,43,62} . Other components explored were personal protective equipment (two studies) ^{38,41} and availability of utilities like running water and electricity (two studies) ^{38,41} .” Page 22 lines 758-760
41	Line 327: “while some personal...” please check this sentence as it seems to run on.	Agreed, we have made changes as follows, “Some PPE especially gloves, were available in most hospitals, while others, such as eye protection, remained unavailable, which is in line with previous research ⁶⁸ .” Page 23 Lines 789-792
42	Line 333: Interesting the authors indicate multilaterals and technical assistance agencies for these priorities. Why not government or decision makers?	Thank you – this is a good point. We have made modifications as follows to emphasize the role of government: “Technical assistance, training of health workers, and improved processes for non ICU based care, development of context-specific critical care guidelines, building capacity in equipment maintenance and biomedical engineering in facilities, and support to medical oxygen infrastructure development should be prioritised by governments and in funding applications to,

		multilaterals and technical assistance agencies 38,67,68” Page 23 lines 794-798
43	Line 346: Unclear here what the authors mean by “eligible studies were inconsistent”. How so? Can detail be provided for why it is not possible to compare or synthesise these results?	Thank you for the question. We meant that the studies reporting costs had variations thus making it difficult to compare. We have made a change as below to make it clearer, “This is further confirmed by the data extracted in this review which showed that reporting of costs within eligible studies varied by defined services provided, breakdowns of costs and inputs, as well as the cost unit, thereby making comparisons or synthesising information across studies difficult.” Page 24 lines 817-820

VERSION 2 – REVIEW

REVIEWER	Morgan Prust Yale University
REVIEW RETURNED	13-Sep-2022
GENERAL COMMENTS	The authors are to be commended for their work on this important topic. They have adequately addressed the concerns I raised in my review of the initial draft. This paper is of relevance for critical care delivery and capacity building not only in Tanzania but in a wide array of other LMICs.